# Non-Cholesterol Sterol Concentrations as Biomarkers for Cholesterol Absorption and Synthesis in Different Metabolic Disorders: A Systematic Review

**DOI:** 10.3390/nu11010124

**Published:** 2019-01-09

**Authors:** Sultan Mashnafi, Jogchum Plat, Ronald P. Mensink, Sabine Baumgartner

**Affiliations:** Department of Nutrition and Movement Sciences, NUTRIM School of Nutrition and Translational Research in Metabolism, Maastricht University Medical Center, 6200 MD Maastricht, The Netherlands; s.mashnafi@maastrichtuniversity.nl (S.M.); j.plat@maastrichtuniversity.nl (J.P.); r.mensink@maastrichtuniversity.nl (R.P.M.)

**Keywords:** non-cholesterol sterols, plant sterols, BMI, diabetes mellitus, metabolic syndrome, hyperlipidemia, cardiovascular disease, intestinal disease, liver disease, kidney disease

## Abstract

Non-cholesterol sterols are validated biomarkers for intestinal cholesterol absorption and endogenous cholesterol synthesis. However, their use in metabolic disturbances has not been systematically explored. Therefore, we conducted a systematic review to provide an overview of non-cholesterol sterols as markers for cholesterol metabolism in different metabolic disorders. Potentially relevant studies were retrieved by a systematic search of three databases in July 2018 and ninety-four human studies were included. Cholesterol-standardized levels of campesterol, sitosterol and cholestanol were collected to reflect cholesterol absorption and those of lathosterol and desmosterol to reflect cholesterol synthesis. Their use as biomarkers was examined in the following metabolic disorders: overweight/obesity (*n* = 16), diabetes mellitus (*n* = 15), metabolic syndrome (*n* = 5), hyperlipidemia (*n* = 11), cardiovascular disease (*n* = 17), and diseases related to intestine (*n* = 16), liver (*n* = 22) or kidney (*n* = 2). In general, markers for cholesterol absorption and synthesis displayed reciprocal patterns, showing that cholesterol metabolism is tightly regulated by the interplay of intestinal absorption and endogenous synthesis. Distinctive patterns for cholesterol absorption or cholesterol synthesis could be identified, suggesting that metabolic disorders can be classified as ‘cholesterol absorbers or cholesterol synthesizers’. Future studies should be performed to confirm or refute these findings and to examine whether this information can be used for targeted (dietary) interventions.

## 1. Introduction

Cholesterol metabolism is tightly regulated, since cholesterol plays an essential role in many physiological processes [1]. Cells have different possibilities to regulate cellular free cholesterol concentrations such as changing low-density lipoprotein-receptor expression, endogenous synthesis, intracellular esterification and excretion [2]. Cholesterol can not only be obtained from de novo endogenous cholesterol synthesis, which occurs virtually in every single cell [3], but also from intestinal absorption of dietary and biliary cholesterol. Interestingly, an inverse relationship exists between cholesterol absorption and synthesis, whereby low intestinal cholesterol absorption is compensated by upregulation of cholesterol synthesis and vice versa [4]. To study the processes of intestinal cholesterol absorption and endogenous cholesterol synthesis in humans, several approaches can be used. Intestinal cholesterol absorption can be measured by using radioisotope tracer or stable isotope tracer methods. Radioisotope methods can be used to quantify cholesterol absorption via both direct and indirect procedures [5]. The only direct method to measure intestinal cholesterol absorption in humans is the intestinal perfusion technique using radioisotopes [6], whereas cholesterol balance and isotope ratio methods can be used to indirectly calculate cholesterol absorption. Cholesterol absorption using stable isotopes can be estimated using the dual plasma isotope ratio method, continuous isotope feeding, or single stable isotopes [5]. For quantifying endogenous cholesterol synthesis, other techniques such as cholesterol balance, fractional conversion of squalene, mass isotopomer distribution analysis (MIDA), and deuterium incorporation (DI) are used. Overall, the cholesterol balance technique, dual plasma isotope ratio, and continuous isotope feeding are the gold standard methods to quantify respectively cholesterol synthesis and absorption. These methods were developed and validated many years ago [7,8,9,10]. However, although these techniques are very precise, they are also complex, laborious, expensive and require a steady-state condition. Thus, these techniques are only suitable for small-scale in-depth studies, but not for large-scale intervention studies [11]. Consequently, there is a clear need for alternative approaches to monitor intestinal cholesterol absorption and endogenous cholesterol synthesis. In the early 90s, serum non-cholesterol sterols were introduced as validated biomarkers for assessing intestinal cholesterol absorption and endogenous synthesis [12]. The cholesterol precursors squalene, desmosterol and lathosterol reflect cholesterol synthesis, and the non-cholesterol sterols sitosterol, campesterol and cholestanol fractional intestinal cholesterol absorption [13].

The use of non-cholesterol sterols as biomarkers to estimate intestinal cholesterol absorption and endogenous cholesterol synthesis has been validated by comparing plasma non-cholesterol sterols with absolute measurements for intestinal cholesterol absorption and whole-body cholesterol synthesis [12]. In more detail, Miettinen and colleagues measured intestinal cholesterol absorption via the dual isotope continuous feeding approach and by determining the ratio of plasma campesterol, sitosterol and cholestanol to total cholesterol (TC) in a randomly selected healthy normocholesterolemic population [12]. It was shown that the ratio of these plasma non-cholesterol sterols to cholesterol highly correlated with the absorption values obtained by using the dual isotope method. In another study, including middle-aged men, plasma cholestanol-to-cholesterol and plant sterol-to-cholesterol ratios were associated with fractional and absolute absorption of dietary cholesterol. From this second study, it was concluded that the serum ratio of cholestanol-to-cholesterol is a sensitive indicator to detect changes in intestinal sterol absorption [14]. Regarding endogenous whole-body cholesterol synthesis, the cholesterol precursors lathosterol, desmosterol and cholestenol, calculated as their ratios to cholesterol, are often used as alternative markers. Their validity was demonstrated by showing positive correlations with endogenous cholesterol synthesis measured by the sterol balance method [12,14,15]. Furthermore, Matthan and colleagues validated the plasma cholesterol precursors squalene, lanosterol, desmosterol and lathosterol by measuring the rate of uptake of deuterium into plasma free cholesterol in a small study with hypercholesterolemic women [16]. The deuterium incorporation method correlated positively with cholesterol synthesis as estimated by the plasma biomarkers. It should be noted that the quality of the methodology used to measure non-cholesterol sterol levels is important for the validity of these biomarkers to reflect cholesterol metabolism [17]. In addition, since cholesterol and non-cholesterol sterol absorption in the intestinal lumen is regulated by the Niemann-Pick C1-like 1 sterol transporter (NPC1L1) and the ABCG5/G8 transporters, genetic variation in these protein transporters may impact their activity, which could possibly have an effect on selectivity of cholesterol absorption biomarkers.

Another approach to address the validity of these biomarkers is by evaluating a change in their levels during drug therapies such as statin and ezetimibe, which have known and accepted effects on cholesterol metabolism [18]. Briefly, statins lower endogenous cholesterol synthesis by inhibiting the rate-limiting enzyme, 3-hydroxy-3-methyl-glutaryl-coenzyme A (HMG-CoA) reductase, leading to reduced production of cholesterol. Ezetimibe inhibits the NPC1L1 sterol transporter in the enterocyte resulting in decreased absorption of dietary and biliary cholesterol [19]. Nissinen and co-workers conducted a clinical trial to evaluate the change in whole-body cholesterol metabolism among hypercholesterolemic men during two different statin treatments and showed that ratios of lathosterol-to-cholesterol and desmosterol-to-cholesterol correlated with absolute cholesterol synthesis measured using isotope tracer techniques at baseline, as well as on statin treatment [20]. This study also demonstrated inverse correlations at baseline between cholesterol synthesis markers (ratios of plasma lathosterol-to-cholesterol and desmosterol-to-cholesterol) and cholesterol absorption markers (ratios of plasma sitosterol-to-cholesterol, campesterol-to-cholesterol and cholestanol-to-cholesterol). In a study by Stellaard et al. [21], ezetimibe treatment of a population with normal daily cholesterol intake, serum ratios of sitosterol-to-cholesterol, campesterol-to-cholesterol and cholestanol-to-cholesterol were shown to reflect changes in fractional cholesterol absorption. In the same study, serum lathosterol-to-cholesterol ratios correlated with absolute cholesterol synthesis. Taking together, findings from validation studies comparing non-cholesterol sterols with radio or stable isotope tracer techniques together with changes in non-cholesterol sterols during pharmacological intervention studies indicate that the use of serum non-cholesterol sterols as biomarkers for cholesterol metabolism are valid under various conditions.

As outlined above, non-cholesterol sterols are used as markers for intestinal cholesterol absorption and endogenous synthesis in healthy subjects under normal metabolic conditions [12,14]. However, there is a need to explore in more detail what we can learn from these non-cholesterol sterols as biomarkers of cholesterol metabolism in various metabolic conditions. Therefore, the aim of this paper is to provide a systematic overview of the use of non-cholesterol sterols as biomarkers to reflect cholesterol metabolism in different metabolic disorders.

## 2. Methods

### 2.1. Search Strategy

Potentially relevant studies were retrieved by a systematic search of three databases (Medline, Embase, and the Cochrane Central Register of Clinical Trials) in July 2018. Search terms consisted of keywords that were related to plant sterols and metabolic conditions. The following search terms were used: plant sterols (plant sterol or plant sterols or phytosterol or phytosterols or sitosterol or campesterol or brassicasterol or stigmasterol or avenasterol or lathosterol or desmosterol or cholestanol or lanosterol or squalene or cholestenol) combined with metabolic syndrome, diabetes, heart disease (cardiovascular disease or cardiovascular diseases or CVD or coronary heart disease or coronary heart diseases or CHD or carotid artery disease or coronary artery disease or coronary artery diseases or CAD or atherosclerosis), hyperlipidemia (hypercholesterolemia or hypercholesterolaemia or hyperlipidemia or hyperlipidaemia or hyperlipoproteinemia or hyperlipoproteinaemia) and organ-related diseases (liver or kidney or kidneys or intestine or intestines or intestinal or small bowel or ileum or ileal or jejunum or duodenum or duodenal or colon or colonic) and plasma (plasma or serum or blood).

### 2.2. Selection of Studies

Human studies investigating cholesterol metabolism using the surrogate non-cholesterol sterol markers in different metabolic conditions were selected. The selection procedures were divided into two stages, and for the first stage, titles and abstracts were screened and papers were included if they met the following criteria: (1) studies performed in human subjects, (2) measurement of plasma (or serum) non-cholesterol sterol concentrations, (3) original research (i.e., no case reports, conference proceedings or reviews), (4) written in English language, and (5) no duplicates. For the second selection stage, full papers were read to assess their eligibility and studies were excluded when they lacked a control group or when plasma TC-standardized non-cholesterol sterols were not reported or could not be calculated. When inconclusive, the eligibility of the studies was discussed with the authors to reach consensus.

### 2.3. Data Collection and Transformation

Data were collected using a spreadsheet that included the following information: publication characteristics (reference number, first author and year of publication), study characteristics (design, specification of subgroups and sample size), subject characteristics (health status, mean age, mean BMI and gender distribution), measurement characteristics (plasma/serum and type of analytical method), and variable characteristics (sitosterol, campesterol, cholestanol, lathosterol, desmosterol and cholesterol). For all variables, mean and variance measures were collected at baseline in the metabolic condition and the control group. If non-cholesterol sterol or TC concentrations were expressed in μg/dL, μg/mL, mg/dL, mg/L, mg/mL, μg/L, or ng/mL, these units were converted to µmol/L and mmol/L based on their molecular weight (sitosterol: 414.7, campesterol: 400.7, lathosterol and cholesterol: 386.7, cholestanol: 388.7, desmosterol: 384.6 g/mol). These conversions applied for means and variance measures. Absolute as well as TC-standardized data were collected, and TC standardized values were calculated if these were not reported. If TC-standardized values were not reported or could not be calculated, the study was excluded. Data reported as median (interquartile range) values were transformed to means and standard deviation (SD) based on the method of Wan et al. [22] and data displayed in graphs were estimated manually. 

## 3. Results

The systematic search retrieved 1953 potentially relevant papers and after two selection rounds, 94 studies were included in the systematic review. A flowchart of the study selection process is presented in Figure 1.

### 3.1. Serum Non-Cholesterol Sterol Markers in Overweight and Obese Subjects

Sixteen articles carried out in overweight or obese subjects met the inclusion criteria (Table 1). In these 16 papers, eight studies were reported that compared cross-sectionally obese subjects with normal weight subjects [23,24,25,26,27,28,29,30], while two intervention studies compared subjects before and after surgery [31,32], and six intervention studies before and after diet-induced weight loss [33,34,35,36,37,38]. The mean BMI (Body Mass Index) in the cross-sectional studies of the cases was >30 kg/m^2^ in six comparisons, between 25–30 kg/m^2^ in one comparison [24], and not reported in another comparison [25]. Control subjects had a mean BMI in the normal-weight range (20.9–25.1 kg/m^2^).

In general, the cross-sectional studies suggested that an increased BMI was associated with a lower cholesterol absorption and higher cholesterol synthesis. Surprisingly, TC-standardized levels of sitosterol and campesterol and of lathosterol both increased in subjects after sleeve gastrectomy [31] or gastric bypass [32]. In contrast, cholesterol absorption was higher and synthesis lower before weight loss induced by gastric banding surgery. The intervention studies also showed that diet-induced weight loss increased cholesterol absorption and decreased cholesterol synthesis.

Overall, these findings suggest cholesterol absorption is decreased and synthesis increased in obese subjects. These associations are reversed after diet-induced weight loss. Effects of the different types of surgical interventions are not clear.

### 3.2. Serum Non-Cholesterol Sterol Markers in Subjects with Diabetes Mellitus

Fifteen studies were identified, of which four studies were performed in patients with type 1 diabetes mellitus (T1DM) [39,40,41,42], and nine studies in patients with type 2 diabetes mellitus (T2DM) of which one paper yielded two data points [43,44,45,46,47,48,49,50,51], while in one study, both T1DM and T2DM patients were included [52] (Table 2). In one study, women with gestational diabetes (GDM) were examined [53].

Three studies in subjects with T1DM suggested that cholesterol absorption was higher in T1DM patients [39,40,42], and four studies reported that cholesterol synthesis was lower in T1DM patients compared to non-diabetic controls [39,40,42,52]. A study performed in children [41] did not find any differences in cholesterol absorption and synthesis markers between cases and controls. The opposite was found for T2DM, i.e., four studies [43,44,45,50] suggested lower cholesterol absorption and higher cholesterol synthesis in T2DM patients. Two of these studies [43,50] were performed in T2DM subjects on statin therapy, but cholesterol absorption and synthesis patterns were comparable to those of T2DM patients not using statins [44,45]. In two studies, no differences in cholesterol absorption markers were observed [47,51], and three studies did not report differences in synthesis markers in T1DM patients compared to non-diabetics [47,49,51,52]. Finally, pregnant women with gestational diabetes showed comparable levels of cholesterol absorption and synthesis markers compared to non-diabetic pregnant women (Data tabulated in Appendix A) [53]. 

To summarize, most studies in T1DM patients suggested that cholesterol absorption is higher and cholesterol synthesis lower compared to non-diabetic control subjects. For T2DM patients, the opposite was found. No effects of GDM were reported.

### 3.3. Serum Non-Cholesterol Sterol Markers in Hyperlipidemic Subjects

Eleven studies met the inclusion criteria (Table 3). Three studies were performed in patients with Familial Combined Hyperlipidemia (FCH) [54,55,56], one study in patients with Familial Hypercholesterolemia (FH) [57], and four studies included patients with FCH, FH or another (non-FH) form of hypercholesterolemia [58,59,60,61]. Patients with hypercholesterolemia other than FCH of FH were evaluated in three studies [62,63,64].

In patients with FCH, cholesterol absorption markers were significantly lower compared to control subjects in four studies [55,56,58,60], comparable in one study [61], and not measured in one study [54]. At least one of the cholesterol synthesis markers was significantly increased in five studies [54,55,56,60,61], and were comparable between FCH patients and control subjects in one study [58]. Two studies in FH patients, of which one was performed in children, reported comparable cholesterol absorption markers between cases and controls subjects, while cholesterol synthesis was lower [58] or comparable between patients with or without FH [57]. 

In patients diagnosed with hypercholesterolemia other than FCH or FH, at least one of the cholesterol absorption markers increased in three comparisons [58,60,61], was comparable in two comparisons [64], and was not tested or measured in three comparisons [59,62,63]. One study reported higher cholesterol absorption in subjects diagnosed with heterogeneous autosomal dominant hypercholesterolemia compared to normolipidemic control subjects [61]. For cholesterol synthesis, most comparisons did not find a difference between hypercholesterolemic cases and their controls [60,61,62,64], while cholesterol synthesis was suggested to be higher in one study [61] and lower in another study [58] in cases compared to controls. Two studies measured cholesterol synthesis, but did report differences between cases and control subjects, although it seems that cholesterol synthesis was lower in subjects with hypercholesterolemia [59,63]. 

Overall, studies in patients with FCH suggested that cholesterol absorption was lower and cholesterol synthesis higher compared to control subjects, while patients with FH showed a comparable cholesterol absorption and synthesis pattern as control subjects. Studies in patients with hypercholesterolemia other than FCH or FH suggested a pattern of increased cholesterol absorption, while cholesterol synthesis seems comparable or decreased compared to normolipidemic subjects.

### 3.4. Serum Non-Cholesterol Sterol Markers in Subjects with the Metabolic Syndrome

Five studies met the inclusion criteria for the metabolic syndrome (Table 4). Findings suggested that cholesterol absorption was lower and synthesis higher in subjects with the metabolic syndrome. It should be noted, however, that in four studies [65,66,67,68], cases were obese and had a BMI >30 kg/m^2^, while controls had a BMI around 25 kg/m^2^. It remains to be elucidated whether the metabolic syndrome per se affects cholesterol absorption or synthesis, or if relations can be explained by one of the underlying characteristics of the metabolic syndrome.

### 3.5. Serum Non-Cholesterol Sterol Markers in Subjects with Cardiovascular Diseases

Seventeen studies matched the inclusion criteria (Table 5). The studies varied widely in the selected patient groups. For many studies, it was reported that at least a part of the population had comorbidities such as T2DM [70,71,72,73,74,75,76,77,78], T1DM [79] or hypertension, dyslipidemia, or the metabolic syndrome [73,76,77,78,79]. In general, controls were to some extent matched for these comorbidities. 

One or more of the cholesterol absorption markers were significantly increased in seven of the comparisons, decreased in two comparisons, not statistically different in eleven comparisons, and not tested in the two remaining comparisons. In one study, no cholesterol absorption markers were measured [76]. At least one of the cholesterol synthesis markers was significantly increased in two comparisons, decreased in five comparisons, not statistically different in nine comparisons, not tested in three comparisons. In two studies, opposite findings were reported, as cholesterol-standardized lathosterol levels increased and those of desmosterol decreased [80,81]. Surprisingly, in these two studies, only postmenopausal women were included. Also, in another study with only postmenopausal women, only a significant increase in desmosterol was reported [82]. 

Overall, studies in patients with cardiovascular suggested a pattern of increased cholesterol absorption. Relationships with cholesterol synthesis markers were less clear. In the majority of studies, however, decreases or no differences were reported. Whether effects on lathosterol and desmosterol are gender-specific warrant further study.

### 3.6. Serum Non-Cholesterol Sterol Markers in Subjects with Intestinal Diseases

Six studies were identified that matched our inclusion criteria, of which one had three study arms, providing eight comparisons versus controls in total (Table 6). Three of these studies evaluated conditions where the length of the small intestine was reduced, i.e., two studies in patients with short bowel syndrome [87,88], and one study in Familial Hypercholesterolemia (FH) patients who underwent ileal bypass surgery [89]. Another study evaluated three conditions where the colon was removed, i.e., ileostomy, ileorectal anastomosis and ileal pouch anastomosis [90]. This latter condition was also studied by Hakala et al. [91]. Finally, there was one study in patients with gastric bleeding [92].

Two studies in subjects with a shortened small intestine suggested that cholesterol synthesis increased [87,88], whereas only one study showed a lower cholesterol absorption [88] as compared to controls. In contrast to the first study by Ellegard et al. [87], the study showing the effects on cholesterol absorption was conducted in children [88]. The third study in subjects with an ileal bypass plus FH did not show any differences versus controls. In two studies including subjects without a colon cholesterol synthesis increased, i.e., in ileostomy patients and patients with ileal pouch anastomosis [90], whereas two other studies did not show a difference in cholesterol synthesis, i.e., in patients with ileorectal anastomosis [90] and ileal pouch anastomosis [91]. In all four comparisons, cholesterol absorption increased [90,91], although this conclusion was not consistent for all absorption markers. Finally, data from subjects with gastric bleeding did not suggest differences in cholesterol synthesis or absorption.

To summarize, studies in patients with shorter small intestines suggested that cholesterol synthesis is higher and cholesterol absorption lower compared to control subjects. For patients without a colon, both cholesterol synthesis and absorption seem higher.

### 3.7. Serum Non-Cholesterol Sterol Markers in Subjects with Kidney Disease

Two studies were identified, of which one study was performed in hyperlipidemic patients with nephrotic proteinuria [93] and one in hemodialysis subjects with diabetes type 2 plus manifestations of CVD [94] (Table 7). Both studies suggested that endogenous synthesis is lower in patients with kidney disease as compared to controls without kidney disease. The study in hemodialysis patients showed a higher cholesterol absorption in patients as compared to controls [94], whereas absorption markers were not reported in the study including patients with nephrotic proteinuria [93]. To summarize, these two studies suggested that cholesterol absorption is higher and cholesterol synthesis lower compared to controls, even in the presence of T2DM.

### 3.8. Serum Non-Cholesterol Sterol Markers in Subjects with Liver Diseases

Twenty-two studies were identified, of which five studies were performed in patients with steatosis [95,96,97,98,99], eleven studies in patients with cholestasis [100,101,102,103,104,105,106,107,108,109,110], and six studies included patients with liver diseases related to cirrhosis or necrosis [111,112,113,114,115,116] (Table 8).

In patients with a fatty liver (steatosis), cholesterol absorption markers were lower in one study [96], comparable to control subjects in another study [97] and not significantly tested or measured in three studies [95,98,99]. At least one of the cholesterol synthesis markers increased in two studies (three comparisons) [95,97], which was independent of statin use, comparable in one study [96] and not significantly tested in two studies [98,99]. 

In nine comparisons in cholestasis patients with gallstones, at least one of the cholesterol absorption markers decreased in three comparisons [106,108] comparable in one [105] and not tested or measured in five comparisons [100,102,104,109]. On the other hand, cholesterol synthesis markers increased in five comparisons [102,107,108] comparable in two [106,109] and not tested or measured in the remaining comparisons. In children with gallstones, results are inconsistent. In patients with black pigment stones, cholesterol absorption was comparable, while cholesterol synthesis increased compared to controls. On the other hand, patients with cholesterol stones had significantly lower cholesterol absorption as well as cholesterol synthesis markers. In cholecystectomized patients, cholesterol absorption was comparable to controls, while cholesterol synthesis increased [101], and in children after successful surgery for biliary atresia, cholesterol absorption markers were inconsistent, while cholesterol synthesis was lower in cases compared with healthy controls [110]. Markers for cholesterol absorption and synthesis were also measured in pregnant women with cholestasis, but differences compared to control subjects were not significantly tested [103].

In patients with primary biliary cirrhosis, markers for cholesterol absorption are inconsistent, i.e., increased concentrations in two comparisons [111,112], reduced concentrations in one comparison [114], and not tested in two comparisons [111,115]. Patients with acute necrosis have a higher cholesterol absorption compared with controls [114], while cholesterol absorption was comparable in children with Intestinal Failure Associated Liver Disease (IFALD) [113]. At least one cholesterol synthesis marker was reduced in primary biliary cirrhosis, Hepatitis C-related cirrhosis and acute necrosis [111,114,116], while cholesterol synthesis increased in IFALD patients [113]. 

To summarize, studies in patients with steatosis or cholestasis suggested a pattern of reduced cholesterol absorption and increased cholesterol synthesis, while it appears that patients with cirrhosis have increased cholesterol absorption and reduced cholesterol synthesis. Studies performed in children with cholestasis and IFALD are inconsistent and warrant further investigation. 

### 3.9. Summary of the Results

An overview of non-cholesterol sterol concentrations as biomarkers for cholesterol absorption and synthesis in different metabolic disorders is presented in Table 9. 

## 4. Conclusions

To better understand cholesterol metabolism in a wide variety of metabolic disorders, we here present the first extensive systematic overview of plasma non-cholesterol sterols levels, which are validated biomarkers for intestinal cholesterol absorption and endogenous cholesterol synthesis, in different metabolic disorders.

In the vast majority of studies, TC-standardized levels of campesterol, sitosterol and cholestanol on the one hand, and TC-standardized levels of lathosterol and desmosterol on the other hand, showed comparable patterns. This underlines their use as biomarkers for cholesterol absorption and cholesterol synthesis, respectively. Moreover, non-cholesterol biomarkers displayed reciprocal patterns, indicating that cholesterol metabolism is tightly regulated by the balance of intestinal absorption and endogenous synthesis. Furthermore, we identified that certain metabolic disorders are characterized by either higher cholesterol absorption or by higher cholesterol synthesis, classifying them as ‘cholesterol absorbers or cholesterol synthesizers’. The identification of metabolic disorders as cholesterol absorbers or synthesizers is important for future (dietary) interventions, since pharmacological and dietary treatments have different underlying mechanisms by which they affect cholesterol metabolism [117,118]. In more detail, cholesterol absorbers would benefit from ezetimibe treatment and plant sterol or stanol consumption, since these interventions inhibit intestinal cholesterol absorption [119,120]. On the other hand, subjects with higher cholesterol synthesis would achieve better cholesterol target levels by using statins, i.e., drugs that reduce endogenous cholesterol synthesis rates [121]. 

In general, cholesterol absorption is decreased and synthesis is increased in overweight and obese subjects, in T2DM patients, and in metabolic syndrome subjects. Since the metabolic syndrome is a constellation of at least three of the following features: visceral obesity, impaired fasting glucose, high triacylglycerol, low high-density lipoprotein cholesterol, and hypertension, it could be speculated that low absorption/high synthesis patterns are (partly) explained by insulin resistance, obesity status, or disturbed lipid profile. However, differences in BMI are most likely not the only factor to explain differences in cholesterol metabolism between metabolic syndrome subjects and controls, since Hernandez-Mijares et al. [69] showed comparable results in weight-matched metabolic syndrome subjects. Interestingly, differences between obese and control subjects in cholesterol absorption and/or synthesis are reversed after diet-induced weight loss, while the effect on cholesterol metabolism after weight loss by different types of surgical interventions are inconsistent. It might be possible that energy balance (or the lack thereof) plays a role in the inconsistent effects on cholesterol homeostasis after surgery-induced weight loss. Future research on the effects of weight loss on cholesterol absorption and synthesis is therefore warranted. As stated, T2DM patients also display a low cholesterol absorption/high cholesterol synthesis pattern, while the opposite is true for T1DM patients, where cholesterol absorption is higher and synthesis is lower than in non-diabetic controls. Insulin resistance might be a contributory factor in the lower cholesterol absorption/higher cholesterol synthesis pattern, since both metabolic syndrome and T2DM patients showed a comparable low absorption/high synthesis pattern, which is in contrast to T1DM patients who are insulin-dependent but can still be insulin sensitive [122]. 

We also investigated the interplay in cholesterol absorption and synthesis in hyperlipidemias and cardiovascular diseases. Both FCH and FH are inherited disorders of cholesterol metabolism. However, FH patients are characterized by high serum cholesterol concentrations, while patients with FCH also exhibit high serum triacylglycerol concentrations. Although studies in FH patients suggested that cholesterol absorption and synthesis pattern are comparable to those of normolipidemic subjects, FCH patients are characterized by a pattern of low cholesterol absorption and high cholesterol synthesis. It has been postulated that patients with FCH have an altered cholesterol synthesis pathway, resulting in higher cholesterol synthesis and thus lower cholesterol absorption [56]. In contrast to FCH, hyperlipidemic patients with a cause other than FH or FCH were characterized by higher cholesterol absorption and lower cholesterol synthesis. Patients with CVD also displayed this reciprocal pattern of high absorption/low synthesis. Indeed, most studies have suggested that higher cholesterol absorption and lower cholesterol synthesis is more atherogenic [72,123,124]. Nevertheless, this is difficult to reconcile with the low absorption/high synthesis pattern in T2DM and FCH patients, who are also at increased risk to develop CVD. The relationship between cholesterol metabolism and CVD risk and the underlying mechanisms should therefore be investigated in more detail in future studies.

In organ-specific diseases (intestine, liver and kidney), the relationship between cholesterol absorption and synthesis was more heterogeneous. In subjects with a deprived small intestine, cholesterol synthesis increased, while cholesterol absorption decreased or comparable to control subjects. Since cholesterol is mainly absorbed at the duodenum and proximal jejunum, the upregulation of cholesterol synthesis is most likely a counter-reaction in patients with a shorter small intestine. On the other hand, in patients without a colon, both cholesterol synthesis and absorption appeared to be higher. This finding does not concur with the notion of reciprocal changes in cholesterol homeostasis and might suggest that absence of microbiota could play a role in cholesterol metabolism in patients with a deprived large intestine. In patients with kidney-related diseases (nephrotic proteinuria or hemodialysis), cholesterol absorption was higher and synthesis was lower. These patients were hyperlipidemic or at risk to develop CVD, which might partly explain the high cholesterol absorption/low cholesterol synthesis pattern. Patients with chronic kidney disease also have a high CVD risk [125] and it might be of interest to investigate whether changes in cholesterol metabolism could play a role in the CVD risk of kidney patients. In patients with steatosis (fatty liver), cholesterol absorption was lower and cholesterol synthesis higher compared with control subjects. Simonen et al. [99] indeed demonstrated that biomarkers of cholesterol synthesis correlated positively with liver fat—independent of BMI—suggesting that cholesterol synthesis increases when the liver contains more fat. In patients with cholestasis, the same pattern is seen, i.e., reduced cholesterol absorption and increased cholesterol synthesis. For patients suffering from a cirrhotic or necrotic liver, it appears that the liver can no longer maintain its cholesterol synthesis, which most likely results in an increased cholesterol absorption. Few studies were performed in children with liver diseases, and with inconsistent results, warranting further investigation.

Overall, distinctive patterns for cholesterol absorption or cholesterol synthesis could be identified, suggesting that metabolic disorders can be classified as ‘cholesterol absorbers or cholesterol synthesizers’. It should be noted that our conclusions are mainly based on cross-sectional studies, which makes it impossible to draw any conclusions on causal relationships. Therefore, future research should confirm or refute our findings. Ultimately, the classification of a metabolic disorder as cholesterol absorber or synthesizer based on non-cholesterol sterol biomarkers can be used for targeted (dietary) interventions.

## Figures and Tables

**Figure 1 nutrients-11-00124-f001:**
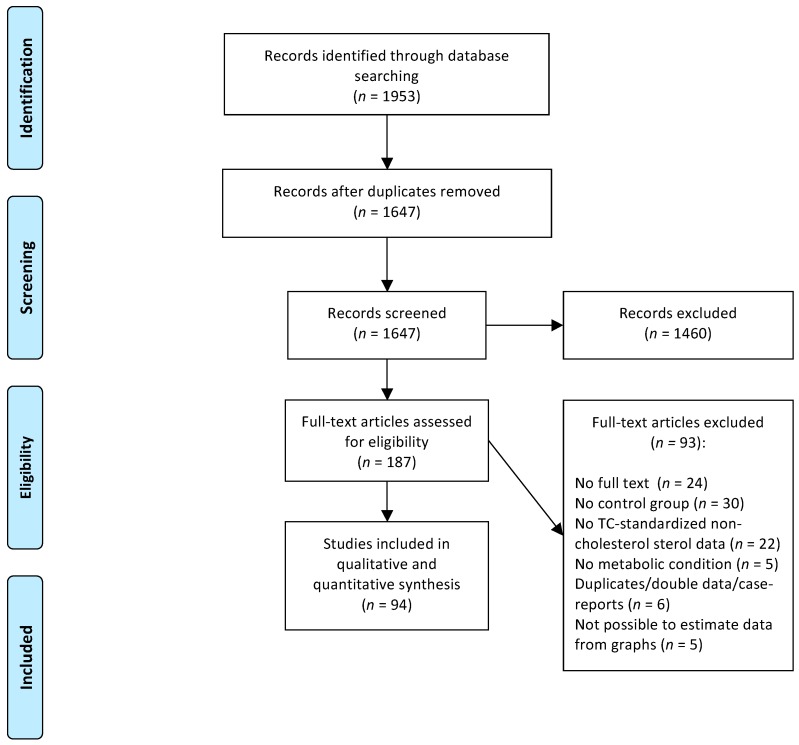
Flowchart of the study selection process.

**Table 1 nutrients-11-00124-t001:** Serum non-cholesterol sterol markers in overweight and obese subjects.

		Sitosterol		Campesterol		Cholestanol		Lathosterol		Desmosterol	
**Cross-Sectional**											
Chan, 2002 [23]	Cases (*n* = 48)							1.87			
	Controls (*n* = 10)							1.54			
Lupattelli, 2012 [24]	Cases (*n* = 63)	0.87 ± 0.43	↓	0.40 ± 0.33	↓			1.04 ± 0.45	↑		
	Controls (*n* = 63)	1.09 ± 0.49		0.56 ± 0.29				0.81 ± 0.35			
Matthan, 2013 [25]	Cases (*n* = 352) ^a^	1.35 ± 0.03		1.87 ± 0.05		1.23 ± 0.02		1.18 ± 0.03		0.57 ± 0.02	
	Cases (*n* = 504) ^b^	1.53 ± 0.03		2.04 ± 0.04		1.21 ± 0.02		1.10 ± 0.02		0.55 ± 0.01	
	Controls (*n* = 603)	1.82 ± 0.03		2.40 ± 0.04		1.19 ± 0.02		1.06 ± 0.02		0.51 ± 0.01	
	Cases (*n* = 331) ^c^	1.45 ± 0.03		1.99 ± 0.05		1.30 ± 0.03		1.15 ± 0.03		0.59 ± 0.02	
	Cases (*n* = 567) ^d^	1.74 ± 0.03		2.34 ± 0.05		1.24 ± 0.02		1.14 ± 0.02		0.59 ± 0.01	
	Controls (*n* = 251)	1.35 ± 0.03		2.39 ± 0.06		1.27 ± 0.03		1.14 ± 0.03		0.62 ± 0.02	
Miettinen, 2000 [26]	Cases (*n* = 10)	0.72 ± 0.04	↓	1.14 ± 0.11	↓	0.64 ± 0.07	↓	1.67 ± 0.18	↑	0.64 ± 0.03	=
	Controls (*n* = 10)	1.29 ± 0.14		1.95 ± 0.18		0.88 ± 0.08		1.17 ± 0.15		0.55 ± 0.01	
Paramsothy, 2011 [27]	Cases (*n* = 37)	1.08 ± 0.43	=	1.68 ± 0.65	=	0.97 ± 0.23	=	1.29 ± 0.55	↑		
	Controls (*n* = 37)	1.27 ± 0.42		1.92 ± 0.64		1.06 ± 0.23		0.95 ± 0.47			
Riches, 1998 [28]	Cases (*n* = 16)							6.47 ± 4.30	↑		
	Controls (*n* = 16)							1.51 ± 1.52			
Simonen, 2002a [29]	Cases (*n* = 44)	0.96 ± 0.05	↓	1.83 ± 0.11	↓	0.89 ± 0.04	↓	2.34 ± 0.12	=	1.16 ± 0.12	↑
	Controls (*n* = 20)	1.22 ± 0.09		2.24 ± 0.17		1.08 ± 0.05		1.96 ± 0.18		0.85 ± 0.05	
Simonen, 2007 [30]	Cases (*n* = 23)	1.02 ± 0.06	↓	1.93 ± 0.16	↓	0.92 ± 0.06	↓	2.51 ± 0.21	↑	1.06 ± 0.10	↑
	Controls (*n* = 10)	1.42 ± 0.13		2.55 ± 0.23		1.17 ± 0.06		1.73 ± 0.09		0.76 ± 0.07	
**Surgery**											
De Vuono, 2017 [31]	Before (*n* = 42)	0.47 ± 0.35	↑	0.49 ± 0.33	=			2.05 ± 0.91	↑		
	After (*n* = 42)	0.42 ± 0.27		0.39 ± 0.33				1.01 ± 0.56			
Pihlajamaki, 2010a [32]	Before (*n* = 29)	0.87 ± 0.33	↑	1.72 ± 0.66	↑	1.32 ± 0.26	=	1.95 ± 0.60	↑	0.97 ± 0.17	↑
	After (*n* = 29)	0.64 ± 0.26		1.24 ± 0.62		1.36 ± 0.27		1.40 ± 0.53		0.80 ± 0.15	
Pihlajamaki, 2010b [32]	Before (*n* = 26)	0.63 ± 0.20	↓	1.25 ± 0.47	↓	1.36 ± 0.36	↓	2.41 ± 0.66	↑	1.33 ± 0.42	↑
	After (*n* = 26)	0.73 ± 0.28		1.56 ± 0.63		1.46 ± 0.36		1.78 ± 0.60		1.12 ± 0.34	
**Diet-induced**											
Chan, 2008 [34]	Before (*n* = 20)							2.90			
	After (*n* = 20)							2.29			
Chan, 2010 [33]	Before (*n* = 10)			0.93				2.07			
	After (*n* = 10)			0.89				1.82			
Mateo-Gallego, 2014 [35]	Before (*n* = 16) ^e^	1.45 ± 0.59	↓	1.56 ± 0.59	=	0.44 ± 0.16	=	1.29 ± 0.45	=	1.50 ± 0.33	=
	After (*n* = 16)	1.62 ± 0.65		1.80 ± 0.69		0.35 ± 0.07		1.31 ± 0.65		1.49 ± 0.31	
	Before (*n* = 34) ^f^	1.18 ± 0.67	↓	1.37 ± 0.59	=	0.45 ± 0.15	=	1.76 ± 0.61	↑	1.84 ± 0.48	=
	After (*n* = 34)	1.25 ± 0.43		1.48 ± 0.43		0.45 ± 0.18		1.34 ± 0.49		1.68 ± 0.40	
Riches, 1999 [36]	Before (*n* = 14)							7.78			
	After (*n* = 14)							6.02			
Simonen, 2000 [37]	Before (*n* = 16)	0.87 ± 0.05	↓	1.62 ± 0.14	↓	0.85 ± 0.04	=	2.26 ± 0.08	=	1.36 ± 0.29	=
	After (*n* = 16)	1.03 ± 0.08		1.97 ± 0.14		0.95 ± 0.05		2.18 ± 0.10		1.09 ± 0.15	
Simonen, 2002b [38]	Before (*n* = 10)	0.84 ± 0.09	=	1.55 ± 0.16	=	0.96 ± 0.04	↓	2.08 ± 0.15	↑	0.82 ± 0.05	=
	After (*n* = 10)	0.79 ± 0.07		1.26 ± 0.11		1.25 ± 0.05		1.59 ± 0.10		0.66 ± 0.04	

Values are mean ± SD and expressed in μmol/mmoL cholesterol. Non-cholesterol markers are significantly lower (↓), higher (↑) or not-significantly different (=) between cases and controls, or not statistically tested (blank). ^a^ Women with BMI >30 or ^b^ between 25–30 kg/m^2^ and ^c^ men with BMI >30 or ^d^ between 25–30 kg/m^2^. ^e^ Subjects with familial hyperlipidemia or with ^f^ familial combined hyperlipidemia.

**Table 2 nutrients-11-00124-t002:** Serum non-cholesterol sterol markers in subjects with diabetes mellitus type 1 or diabetes mellitus type 2.

		Sitosterol		Campesterol		Cholestanol		Lathosterol		Desmosterol	
**T1DM**											
Feillet, 1994 [52]	Cases (*n* = 10)							0.86 ± 0.11	↓		
	Controls (*n* = 10)							1.16 ± 0.07			
Gylling, 2004 [40]	Cases (*n* = 7)	1.94 ± 0.36	=	3.48 ± 0.58	=	1.55 ± 0.15	↑	0.56 ± 0.07	↓		
	Controls (*n* = 5)	1.24 ± 0.14		2.20 ± 0.29		0.92 ± 0.09		0.86 ± 0.12			
Gylling, 2007 [39]	Cases (*n* = 56)	1.88 ± 0.08	=	3.96 ± 0.20	↑	1.64 ± 0.04	=	1.57 ± 0.07	=	0.76 ± 0.02	=
	Controls (*n* = 18)	1.66 ± 0.14		3.10 ± 0.25		1.45 ± 0.09		1.71 ± 0.14		0.84 ± 0.04	
Jarvisalo, 2006 [41]	Cases (*n* = 48)	2.07 ± 0.10	=	4.45± 0.27	=	1.80 ± 0.05	=	1.19 ± 0.05	=	0.74 ± 0.01	=
	Controls (*n* = 79)	1.91 ± 0.08		3.73 ± 0.18		1.77 ± 0.04		1.29 ± 0.05		0.72 ± 0.01	
Kojima, 1999 [42]	Cases (*n* = 12) ^a^	2.33 ± 0.25	↑	3.84 ± 0.44	↑						
	Cases (*n* = 10) ^b^	1.44 ± 0.14		2.63 ± 0.15							
	Controls (*n* = 10)	1.38 ± 0.06		2.02 ± 0.20							
**T2DM**											
Blaha, 2006 [43]	Cases (*n* = 30)	0.67 ± 0.26	↓	0.83 ± 0.66	↓			1.45 ± 0.62	↑		
	Controls (*n* = 30)	1.92 ± 1.49		2.10 ± 1.31				0.88 ± 0.58			
Feillet, 1994 [52]	Cases (*n* = 9)							0.98 ± 0.10	=		
	Controls (*n* = 10)							1.16 ± 0.07			
Gylling, 2010 [44]	Cases (*n* = 76)	1.13 ± 0.07	↓	2.32 ± 0.13	↓	1.37 ± 0.03	↓	1.52 ± 0.08	↑	0.99 ± 0.03	↑
	Controls (*n* = 549)	1.15 ± 0.07		2.78 ± 0.05		1.43 ± 0.01		1.46 ± 0.02		0.90 ± 0.01	
Gylling, 1997 [45]	Cases (*n* = 13)	1.12 ± 0.11	↓	2.04 ± 0.23	↓	0.92 ± 0.05	↓	2.0 ± 0.10	=	1.08 ± 0.05	↑
	Controls (*n* = 18)	1.78 ± 0.11		3.89 ± 0.23		1.17 ± 0.04		1.80 ± 0.11		0.73 ± 0.01	
Kurano, 2018 [46]	Cases (*n* = 46)	3.02		1.82							
	Controls (*n* = 178)	2.17		1.59							
Lau, 2005 ^c^ [47]	Cases (*n* = 14)	0.45 ± 0.10	=	1.31 ± 0.30	=						
	Cases (*n* = 14)	0.45 ± 0.10	=	1.10 ± 0.20	=						
	Controls (*n* = 15)	0.71 ± 0.30		1.29 ± 0.30							
	Controls (*n* = 15)	0.55 ± 0.10		1.43 ± 0.30							
Okada, 2010 [48]	Cases (*n* = 42)	1.16		2.09		1.38		1.52			
	Controls (*n* = 21)	1.13		2.09		1.41		1.25			
Smahelova, 2005 [50]	Cases (*n* = 63)	1.00 ± 0.57	↓	1.44 ± 1.56	↓			1.50 ± 1.19	↑		
	Controls (*n* = 72)	1.70 ± 1.10		1.88 ± 1.13				0.95 ± 0.53			
Smahelova, 2007 [49]	Cases (*n* = 63)	0.94		1.31				1.53			
	Controls (*n* = 72)	1.75		1.84				0.96			
Yoshida, 2006 [51]	Cases (*n* = 13)	0.82 ± 0.13	=					0.91 ± 0.11	=		
	Controls (*n* = 16)	0.73 ± 0.10						0.73 ± 0.07			

Values are mean ± SD and expressed in μmol/mmoL cholesterol. Non-cholesterol markers are significantly lower (↓), higher (↑) or not-significantly different (=) between cases and controls, or not statistically tested (blank). T1DM: diabetes mellitus type 1, T2DM: diabetes mellitus type 2. ^a^ Subjects on conventional insulin therapy or ^b^ intensive insulin therapy. ^c^ Baseline levels of a randomized clinical trial with 4 arms.

**Table 3 nutrients-11-00124-t003:** Serum non-cholesterol sterol markers in hyperlipidemic subjects.

		Sitosterol		Campesterol		Cholestanol		Lathosterol		Desmosterol	
**FCH**											
Baila-Rueda, 2014 [54]	Cases (*n* = 107)									6.65	↑
	Controls (*n* = 126)									4.14	
Brouwers, 2013 [55]	Cases (*n* = 103)	1.14 ± 0.53	↓	1.82 ± 0.80	↓	1.34 ± 0.48	↓	1.79 ± 0.61	↑	0.38 ± 0.18	=
	Controls (*n* = 240)	1.30 ± 0.53		2.70 ± 0.81		1.60 ± 0.53		1.50 ± 0.55		0.35 ± 0.16	
Garcia-Otin, 2007 [58]	Cases (*n* = 38)	2.17 ± 0.51	↓	2.86 ± 0.62	↓			1.58 ± 0.45	=		
	Controls (*n* = 45)	2.68 ± 0.57		3.37 ± 0.67				1.68 ± 0.44			
Lupattelli, 2012 [60]	Cases (*n* = 38)	0.81 ± 0.33	=	0.46 ± 0.58	↓			1.25 ± 0.61	↑		
	Controls (*n* = 19)	0.95 ± 0.37		0.82 ± 0.59				0.88 ± 0.49			
Noto, 2010 [61]	Cases (*n* = 29)	1.80 ± 0.80	=	1.98 ± 0.90	=			0.69 ± 0.30	↑		
	Controls (*n* = 79)	1.50 ± 0.70		1.82 ± 0.80				0.50 ± 0.30			
Van Himbergen, 2010 [56]	Cases (*n* = 103)	1.14 ± 0.53	↓	1.82 ± 0.80	↓	1.34 ± 0.48	↓	1.79 ± 0.61	↑	0.38 ± 0.19	=
	Controls (*n* = 204)	1.30 ± 0.50		2.07 ± 0.81		1.60 ± 0.53		1.50 ± 0.55		0.35 ± 0.16	
**FH**											
Hirayama, 2013 [59]	Cases (*n* = 47)	0.50		1.79				1.08		0.19	
	Controls (*n* = 32)	0.50		1.55				1.34		0.38	
Ketomaki, 2003 [57]	Cases (*n* = 18)	1.76 ± 0.14	=	3.30 ± 0.35	=	1.57 ± 0.05	=	1.02 ± 0.07	=	0.60 ± 0.02	=
	Controls (*n* = 29)	1.77 ± 0.11		3.58 ± 0.19		1.51 ± 0.05		1.08 ± 0.06		0.55 ± 0.02	
Garcia-Otin, 2007 [58]	Cases (*n* = 31)	3.08 ± 0.70	=	4.00 ± 0.88	=			1.19 ± 0.33	↓		
	Controls (*n* = 45)	2.68 ± 0.57		3.37 ± 0.67				1.68 ± 0.44			
**Non-FH**											
Baila-Rueda, 2017	Cases (*n* = 200)									2.53	=
	Controls (*n* = 100)									2.41	
Garcia-Otin, 2007 [62]	Cases (*n* = 21)	3.93 ± 1.14	↑	5.10 ± 1.40	↑			1.00 ± 0.36	↓		
	Controls (*n* = 45)	2.68 ± 0.57		3.37 ± 0.67				1.68 ± 0.44			
Hirayama, 2013 [59]	Cases (*n* = 47)	0.50		1.79				1.08		0.19	
	Controls (*n* = 32)	0.50		1.55				1.34		0.38	
Lupattelli, 2012 [60]	Cases (*n* = 53)	1.30 ± 0.81	↑	0.79 ± 0.81	=			0.95 ± 0.68	=		
	Controls (*n* = 19)	0.95 ± 0.37		0.82 ± 0.59				0.88 ± 0.49			
Nagy, 2006 [63]	Cases (*n* = 6)	0.81				3.09		0.52		0.14	
	Controls (*n* = 6)	0.68				3.20		0.62		0.21	
Noto, 2010 [61]	Cases (*n* = 19) ^a^	1.20 ± 0.70	↓	0.83 ± 0.50	↓			1.04 ± 0.50	↑		
	Cases (*n* = 2) ^b^	1.54 ± 2.01	=	0.66 ± 1.15	=			0.95 ± 3.40	=		
	Cases (*n* = 63) ^c^	1.97 ± 0.70	↑	2.04 ± 0.70	↑			0.49 ± 0.30	=		
	Controls (*n* = 79)	1.50 ± 0.70		1.82 ± 0.80				0.50 ± 0.30			
Von Bergmann, 2003 [64]	Cases (*n* = 8) ^d^			1.97 ± 0.29	=	2.88 ± 0.80	=	1.64 ± 0.24	=		
	Cases (*n* = 6) ^e^			1.92 ± 0.21	=	2.72 ± 0.40	=	2.03 ± 0.40	=		
	Controls (*n* = 6)			2.41 ± 0.18		2.74 ± 0.40		2.39 ± 0.58			

Values are mean ± SD and expressed in μmol/mmol cholesterol. In case absolute concentrations were reported, only TC-standardized means were calculated. Non-cholesterol markers are lower (↓) or higher (↑) in cases compared to control, comparable between cases and controls (=) or not statistically tested (blank). FCH: Familial Combined Hyperlipidemia, FH: Familial Hypercholesterolemia. ^a^ Subjects with heterozygous or ^b^ homozygous autosomal dominant hypercholesterolemia or ^c^ polygenic hypercholesterolemia. ^d^ Subjects carrier for apolipoprotein E2/2 or for ^e^ apolipoprotein E4/4.

**Table 4 nutrients-11-00124-t004:** Serum non-cholesterol sterol markers in subjects with the metabolic syndrome.

		Sitosterol		Campesterol		Cholestanol		Lathosterol		Desmosterol	
Chan, 2003 [65]	Cases (*n* = 35)			2.63 ± 0.19	↓			2.58 ± 0.15	↑		
	Controls (*n* = 9)			3.28 ± 0.32				1.90 ± 0.31			
Chan, 2003 [66]	Cases (*n* = 35)			2.60	↓			2.60	↑		
	Controls (*n* = 10)			3.40				1.90			
Gylling, 2007 [67]	Cases (*n* = 74)	0.97 ± 0.05	↓	1.84 ± 0.09	↓	1.19 ± 0.03	↓	2.02 ± 0.07	↑	0.79 ± 0.04	↑
	Controls (*n* = 74)	1.25 ± 0.06		1.84 ± 0.12		1.58 ± 0.04		1.33 ± 0.05		0.82 ± 0.02	
Hernandez-Mijares, 2011 [69]	Cases (*n* = 24)	1.14		0.81							
	Controls (*n* = 24)	1.60		1.15							
Ooi, 2009 [68]	Cases (*n* = 140)			1.59 ± 0.09	↓			2.13 ± 0.08	↑		
	Controls (*n* = 10)			3.17 ± 0.30				1.51 ±0.22			

Values are mean ± SD and expressed in μmol/mmol cholesterol. Non-cholesterol markers are significantly lower (↓), higher (↑) or not-significantly different (=) between cases and controls, or not statistically tested (blank).

**Table 5 nutrients-11-00124-t005:** Serum non-cholesterol sterol markers in subjects with cardiovascular diseases.

		Sitosterol		Campesterol		Cholestanol		Lathosterol		Desmosterol	
Assmann, 2006 [83]	Cases (*n* = 159)	0.75		1.53				0.68			
	Controls (*n* = 318)	0.72		1.53				0.72			
Escurriol, 2010 [84]	Cases (*n* = 299)	1.32 ± 0.52	=	1.53 ± 0.57	=			1.61 ± 0.61	=		
	Controls (*n* = 584)	1.39 ± 0.55		1.56 ± 0.58				1.59 ± 0.68			
Fassbender, 2008 [70]	Cases (*n* = 58)	1.27 ± 0.720	=	1.39 ± 0.86	=			1.33 ± 0.86	=	1.17 ± 1.73	=
	Controls (*n* = 957)	1.42 ± 0.66		1.54 ± 0.77				1.17 ± 0.50		0.84 ± 0.36	
Gylling, 2006 [82]	Cases (*n* = 22)	1.47 ± 0.13	=	3.01 ± 0.31	=	1.27 ± 0.09	=	1.70 ± 0.12	=	0.91 ± 0.07	↑
	Controls (*n* = 14)	1.44 ± 0.09		2.74 ± 0.21		1.19 ± 0.06		1.75 ± 0.12		0.62 ± 0.03	
Gylling, 2009 [80]	Cases (*n* = 47)	1.56 ± 0.10	↑	2.87 ± 0.12	↑	1.30 ± 0.06	=	1.68 ± 0.08	↓	0.96 ± 0.08	↑
	Controls (*n* = 62)	1.24 ± 0.06		2.27 ± 0.11		1.29 ± 0.04		1.96 ± 0.07		0.76 ± 0.02	
Gylling, 1996 [71]	Cases (*n* = 7)	1.39 ± 0.10	↑	3.00 ± 0.21	↑	1.02 ± 0.07	=	2.05 ± 0.22	=	1.01 ± 0.07	=
	Controls (*n* = 6)	0.80 ± 0.09		1.63 ± 0.14		0.83 ± 0.08		2.00 ± 0.26		1.16 ± 0.07	
Matthan, 2009 [72]	Cases (*n* = 155)	1.69 ± 0.06	↑	2.29 ± 0.07	↑	1.44 ± 0.05	↑	1.16 ± 0.04	↓	0.73 ± 0.03	↓
	Controls (*n* = 414)	1.49 ± 0.03		1.96 ± 0.04		1.35 ± 0.03		1.38 ± 0.03		0.75 ± 0.02	
Mori, 2017 [73]	Cases (*n* = 103) ^a^			3.48 ± 1.25	↑			0.61 ± 0.33	↑		
	Controls (*n* = 40)			2.44 ± 1.42				0.44 ± 0.16			
	Cases (*n* = 42) ^b^			2.89 ± 0.94	=			1.28 ± 0.87	=		
	Controls (*n* = 42)			2.83 ± 0.97				1.45 ± 0.53			
Nasu, 2013 [74]	Cases (*n* = 42)	1.46 ± 0.43	↑	2.00 ± 0.37	↑			0.61 ± 0.18	↓		
	Controls (*n* = 38)	1.13 ± 0.32		1.44 ± 0.52				0.97 ± 0.66			
Pinedo, 2007 [75]	Cases (*n* = 373)	1.22 ± 0.47	↓	1.86 ± 0.79	=			1.21 ± 0.47	=		
	Controls (*n* = 758)	1.31 ± 0.46		1.94 ± 0.80				1.18 ± 0.48			
Rajaratnam, 2000 [81]	Cases (*n* = 48)	1.52 ± 0.09	↑	2.78 ± 0.19	↑	1.30 ± 0.06	=	1.71 ± 0.08	↓	0.98 ± 0.09	↑
	Controls (*n* = 61)	1.25 ± 0.06		2.28 ± 0.11		1.30 ± 0.04		1.96 ± 0.07		0.76 ± 0.02	
Shay, 2009 [79]	Cases (*n* = 82)	1.35 ± 0.67	=	2.13 ± 1.15	=			0.47 ± 0.30	↓	0.34 ± 0.20	↓
	Controls (*n* = 213)	1.45 ± 0.76		2.29 ± 1.12				0.54 ± 0.38		0.42 ± 0.32	
Sonoda, 1992 [76]	Cases (*n* = 22)							2.64			
	Controls (*n* = 33)							2.13			
Weingartner, 2009 [77]	Cases (*n* = 8) ^c^			1.47 ± 0.66	=			0.60 ± 0.23	↓		
	Cases (*n* = 4) ^d^			1.51 ± 0.93	=			0.77 ± 0.24	=		
	Cases (*n* = 6) ^e^			1.47 ± 0.69	=			0.87 ± 0.24	=		
	Controls (*n* = 22)			1.39 ± 0.61				1.12 ± 0.47			
Weingartner, 2011 [85]	Cases (*n* = 66)	1.35 ± 0.41	=	1.82 ± 1.04	↑			1.26 ± 0.62	↓		
	Controls (*n* = 111)	1.21 ± 0.03		1.50 ± 0.69				1.38 ± 0.63			
Wilund, 2004 [86]	Cases (*n* = 323) ^f^	0.76 ± 0.54	=	1.27 ± 0.77	=						
	Controls (*n* = 808)	0.78 ± 0.50		1.39 ± 0.78							
	Cases (*n* = 209) ^g^	0.73 ± 0.58	=	1.20 ± 0.81	=						
	Controls (*n* = 1202)	0.76 ± 0.51		1.28 ± 0.74							
Windler, 2009 [78]	Cases (*n* = 186)	0.63		1.04				0.58			
	Controls (*n* = 231)	0.68		1.13				0.80			

Values are mean ± SD and expressed in μmol/mmol cholesterol. Non-cholesterol markers are significantly lower (↓), higher (↑) or not-significantly different (=) between cases and controls, or not statistically tested (blank). ^a^ Subjects with or ^b^ without statin therapy. ^c^ Subjects with three-vessel disease, ^d^ two-vessel disease or ^e^ single-vessel disease. ^f^ Study performed in men or ^g^ women.

**Table 6 nutrients-11-00124-t006:** Serum non-cholesterol sterol markers in subjects with intestinal diseases.

		Sitosterol		Campesterol		Cholestanol		Lathosterol		Desmosterol	
**DSI**											
Ellegard, 2005 [87]	Cases (*n* = 16)							8.08 ± 2.73	↑		
	Controls (*n* = 21)							2.28 ± 0.70			
Pakarinen, 2010 ^c^ [88]	Cases (*n* = 12)	1.95 ± 1.24	=	2.54 ± 2.00	↓			3.53 ± 1.18	↑	1.41 ± 0.20	↑
	Controls (*n* = 80)	1.96 ± 0.50		3.95 ± 0.95				1.15 ± 0.19		0.69 ± 0.14	
Vanhanen, 1992 [89]	Cases (*n* = 6)	1.52	=	2.30	=	1.00	=	1.27	=	1.00	=
	Controls (*n* = 7)	1.40		2.10		1.15		1.23		0.55	
**DLI**											
Hakala, 1997 [91]	Cases (*n* = 12)	1.83 ± 0.12	=	3.16 ± 0.26	↑	1.49 ± 0.08	=	1.90 ± 0.19	=	0.70 ± 0.05	=
	Controls (*n* = 10)	1.44 ± 0.20		1.80 ± 0.29		1.48 ± 0.10		2.03 ± 0.13		0.76 ± 0.08	
Nissinen, 2004 [90]	Cases (*n* = 34) ^a^	1.56 ± 0.09	↑	3.51 ± 0.17	↑	1.40 ± 0.05	=	2.30 ± 0.14	↑	0.91 ± 0.06	=
	Cases (*n* = 8) ^b^	1.39 ± 0.18	=	2.66 ± 0.33	↑	1.18 ± 0.05	↓	3.37 ± 0.75	↑	1.04 ± 0.21	=
	Cases (*n* = 6) ^c^	1.98 ± 0.19	↑	4.03 ± 0.48	↑	1.35 ± 0.09	=	1.69 ± 0.22	=	0.75 ± 0.05	=
	Controls (*n* = 29)	1.27 ± 0.09		1.85 ± 0.16		1.45 ± 0.06		1.88 ± 0.09		0.74 ± 0.03	
**Bleeding**											
Hrabovsky, 2012 [92]	Cases (*n* = 24)	1.13		2.19				1.11			
	Controls (*n* = 100)	0.99		2.04				1.37			

Values are mean ± SD and expressed in μmol/mmol cholesterol. In case absolute concentrations were reported, only TC-standardized means were calculated. Non-cholesterol markers are lower (↓) or higher (↑) in cases compared to control, comparable between cases and controls (=) or not statistically tested (blank). DSI: Deprived Small Intestine, DLI: Deprived Large Intestine. ^a^ Subjects underwent ileal pouch anastomosis, ^b^ conventional ileostomy or ^c^ ileorectal anastomosis.

**Table 7 nutrients-11-00124-t007:** Serum non-cholesterol sterol markers in subjects with kidney disease.

		Sitosterol	Campesterol		Cholestanol		Lathosterol		Desmosterol
Dullaart, 1996 [93]	Cases (*n* = 11)						0.99 ± 0.43	↓	
	Controls (*n* = 22)						1.29 ± 0.41		
Rogacev, 2012 [94]	Cases (*n* = 113)				2.40	↑	1.25	↓	
	Controls (*n* = 229)				1.25		1.40		

Values are mean ± SD and expressed in μmol/mmoL cholesterol. In case absolute concentrations were reported, only TC-standardized means were calculated. Non-cholesterol markers are lower (↓) or higher (↑) in cases compared to control, comparable between cases and controls (=) or not statistically tested (blank).

**Table 8 nutrients-11-00124-t008:** Serum non-cholesterol sterol markers in subjects with liver diseases.

		Sitosterol		Campesterol		Cholestanol		Lathosterol		Desmosterol	
**Steatosis**											
Brindisi, 2012 [95]	Cases (*n* = 74) ^a^	1.26		1.60				1.81 ± 1.30	↑		
	Controls (*n* = 63)	1.30		1.65				1.31 ± 0.80			
	Cases (*n* = 91) ^b^	0.90		1.29				3.50 ± 1.49	↑		
	Controls (*n* = 35)	1.05		1.34				2.72 ± 1.24			
Ikegami, 2012 [96]	Cases (*n* = 15)	0.73 ± 0.06	↓	0.75 ± 0.07	↓			3.03 ± 0.31	=		
	Controls (*n* = 36)	1.90 ± 0.08		2.34 ± 0.11				3.29 ± 0.17			
Min, 2012 [97]	Cases (*n* = 20)	0.85	=							0.98	↑
	Controls (*n* = 6)	1.35								0.52	
Simonen, 2011 [99]	Cases (*n* = 114)	0.81		1.68		1.19		1.77		0.95	
	Controls (*n* = 128)	1.05		2.22		1.32		1.46		0.83	
Simonen, 2013 [98]	Cases (*n* = 17) ^c^							1.81		0.85	
	Cases (*n* = 23) ^d^							1.55		0.94	
	Controls (*n* = 32)							1.53		0.71	
**Cholestasis**											
Castro, 2007 [100]	Cases (*n* = 18)							0.80			
	Controls (*n* = 9)							0.74			
Galman, 2004 ^e^ [102]	Cases (*n* = 45)							3.00 ± 1.10	↑		
	Controls (*n* = 80)							2.40 ± 0.90			
	Cases (*n* = 20)							2.80 ± 1.20			
	Controls (*n* = 20)							3.00 ± 1.10			
Galman, 2011 [101]	Cases (*n* = 18)			6.90 ± 5.02	=			3.09 ± 1.33	↑		
	Controls (*n* = 222)			8.04 ± 4.23				2.54 ± 0.94			
Gylling, 1998 ^f^ [103]	Cases (*n* = 20)					3.52 ± 0.23		1.95		0.80	
	Cases (*n* = 19)					3.84 ± 0.33		2.30		0.82	
	Controls (*n* = 20)					2.82 ± 0.11		1.90		0.87	
Hillebrant, 2002 [104]	Cases (*n* = 19)							3.2 ± 0.40			
	Controls (*n* = 20)							2.8 ± 0.40			
Jiang, 2009 [105]	Cases (*n* = 12)	4.10	=	1.90	=						
	Controls (*n* = 31)	1.80		2.20							
Kakela, 2017 [106]	Cases (*n* = 95)	0.59 ± 0.24	↓	1.25 ± 0.54	↓	1.50 ± 0.73	=	1.94 ± 0.79	=	1.11 ± 1.01	=
	Controls (*n* = 147)	0.85 ± 0.45		1.78 ± 0.85		1.54 ± 0.34		1.73 ± 0.82		0.86 ± 0.32	
Koivusalo, 2015 [107]	Cases (*n* = 17) ^g^	1.70 ± 0.18	=	3.12 ± 0.34	=	1.86 ± 0.09	↑	1.37 ± 0.17	↑	1.07 ± 0.07	↑
	Cases (*n* = 11) ^h^	1.17 ± 0.22	↓	2.07 ± 0.41	↓	1.26 ± 0.11	↓	1.68 ± 0.21	↑	1.04 ± 0.08	=
	Controls (*n* = 82)	1.75 ± 0.08		3.10 ± 0.15		1.68 ± 0.04		0.87 ± 0.08		0.86 ± 0.03	
Krawczyk, 2012 [108]	Cases (*n* = 112) ^e^	1.07 ± 0.50	↓	1.43 ± 0.80	=			1.28 ± 0.70	↑	0.87 ± 0.50	=
	Controls (*n* = 152)	1.20 ± 0.60		1.58 ± 0.90				1.11 ± 0.78		0.87 ± 0.50	
	Cases (*n* = 100)	0.69 ± 0.38	↓	1.03 ± 0.39	=			1.33 ± 0.52	↑	0.56 ± 0.27	↑
	Controls (*n* = 100)	0.83 ± 0.54		1.01 ± 0.57				1.05 ± 0.43		0.46 ± 0.21	
Muhrbeck, 2017 [109]	Cases (*n* = 41)							0.80 ± 0.08	=		
	Controls (*n* = 72)							0.70 ± 0.05			
Pakarinen, 2010 [110]	Cases (*n* = 17)	1.95 ± 0.77	=	3.00 ± 1.41	↓	2.75 ± 0.81	↑	0.81 ± 0.40	↓		
	Controls (*n* = 129)	1.96 ± 0.51		3.95 ± 0.95		1.60 ± 0.35		1.21 ± 0.32			
**Cirrhosis-Necrosis**											
Del Puppo, 1998 [111]	Cases (*n* = 12) ^i^	5.15 ± 1.17		2.20 ± 0.52				0.64 ± 0.09			
	Controls (*n* = 10)	1.86 ± 0.44		1.34 ± 0.24				0.70 ± 0.08		0.14	
	Cases (*n* = 13) ^j^	5.49 ± 0.92	↑	2.69 ± 0.79	↑			0.67 ± 0.07	↓	0.21	
	Controls (*n* = 10)	1.66 ± 0.44		0.88 ± 0.06				1.05 ± 0.17			
Gylling, 1996 [112]	Cases (*n* = 16)	3.72 ± 0.95	↑	4.62 ± 0.88	↑	4.53 ± 0.79	↑	1.11 ± 0.12	=	0.61 ± 0.08	=
	Controls (*n* = 36) ^k^	1.03 ± 0.06		1.58 ± 0.09		0.82 ± 0.05		1.32 ± 0.08		0.55 ± 0.02	
	Controls (*n* = 8) ^l^	1.05 ± 0.14		1.69 ± 0.31		0.80 ± 0.05		1.40 ± 0.16		0.57 ± 0.10	
Mutanen, 2014 [113]	Cases (*n* = 34)	1.55 ± 1.11	=	2.73 ± 1.83	=	1.79 ± 0.39	=	2.90 ± 2.22	↑	1.22± 0.44	↑
	Controls (*n* = 86)	1.68 ± 0.55		3.06 ± 0.79		1.64 ± 0.29		0.83 ± 0.37		0.85 ± 0.14	
Nikkila, 2005 [115]	Cases (*n* = 67)	10.52 ± 19.34		7.63 ± 13.43		12.95 ± 15.20		1.15 ± 1.63			
	Controls (*n* = 59)	1.96 ± 1.66		3.38 ± 3.57		1.34 ± 0.85		1.96 ± 1.66			
Nikkila, 1992 [114]	Cases (*n* = 8) ^m^	3.20		1.50	↓			0.49	↓	0.50	=
	Cases (*n* = 3) ^n^	2.50	↑	2.80	=			0.62	↓	0.60	=
	Controls (*n* = 27)	1.20		2.10				1.25		0.50	
Ydreborg, 2014 ^o^ [116]	Cases (*n* = 36)							0.94 ± 0.54	↓		
	Controls (*n* = 242)							1.18 ± 0.47			
	Cases (*n* = 8)							0.80 ± 0.63	↓		
	Controls (*n* = 75)							1.19 ± 0.53			

Values are mean ± SD and expressed in μmol/mmol cholesterol. In case absolute concentrations were reported, only TC-standardized means were calculated. Non-cholesterol markers are lower (↓) or higher (↑) in cases compared to control, comparable between cases and controls (=) or not statistically tested (blank). ^a^ Subjects with or ^b^ without statin therapy. ^c^ Subjects with NAFLD or ^d^ NASH. ^e^ Study performed in different ethnicities. ^f^ Randomized clinical trial with three arms in pregnant women with cholestasis. ^g^ Subjects with black pigment stones or ^h^ cholesterol stones. ^i^ Hypercholesterolemic or ^j^ normocholesterolemic subjects. ^k^ Study performed in men or ^l^ women. ^m^ Subjects with liver cirrhosis or ^n^ acute liver necrosis. ^o^ Study performed in two different groups of patients with liver cirrhosis.

**Table 9 nutrients-11-00124-t009:** Serum non-cholesterol sterol markers in different metabolic disorders.

	Cholesterol Absorption Compared to Control	Cholesterol Synthesis Compared to Control
BMI	↓	↑
Diabetes mellitus		
T1DM	↑	↓
T2DM	↓	↑
Hyperlipidemia		
FCH	↓	↑
FH	-	-
Non-FH hyperlipemia	↑	↓
Metabolic Syndrome	↓	↑
CVD	↑	↓
Intestine		
Deprived small intestine	↓	↑
Deprived large intestine	↑	↑
Liver		
Steatosis	↓	↑
Cholestasis	↓	↑
Cirrhosis-necrosis	↑	↓
Kidney	↑	↓

BMI: body mass index, T1DM: diabetes mellitus type 1, T2DM: diabetes mellitus type 2, FCH: familial combined hyperlipidemia, FH: familial hypercholesterolemia, CVD: cardiovascular disease ↓ lower compared to control ↑ higher compared to control.

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
