# Peer review of "Non-Cholesterol Sterol Concentrations as Biomarkers for Cholesterol Absorption and Synthesis in Different Metabolic Disorders: A Systematic Review"

_nutrients, 2019, doi:10.3390/nu11010124_

Reviewer 1 Report

Dear Editor,

After carefully reading the manuscript by Mashnafi and co-workers, I can say that it is a really well-written and interesting paper. No more comments on it.

Author Response

Reviewer #1

After carefully reading the manuscript by Mashnafi and co-workers, I can say that it is a really well written and interesting paper. No more comments on it.

Answer:Thank you for the positive feedback on our paper.

Reviewer 2 Report

To:

Editorial Board

Nutrients

Title: “Non-Cholesterol Sterol Concentrations as Biomarkers for Cholesterol Absorption and Synthesis in Different Metabolic Disorders: A Systematic Review”

Dear Editor,

I read this manuscript and I think that:

- The role of pharmacological treatments can influence the final results of this review as they can interfere with NCS concentrations. The authors should discuss such a point in a dedicated limitation section.

- Furthermore, the influence of diet should also be considered. Scicchitano P et al. (Journal of Functional Foods 2014;6:11-32) outlined the influence of nutraceuticals on lipid levels. Therefore, the need for a discussion about this point should also be targeted. Please provide.

- The authors should include a figure able to represent the entire results of this review in order to make it clearer to the Readers.

Author Response

Reviewer #2

The role of pharmacological treatments can influence the final results of this review as they can interfere with NCS concentrations. The authors should discuss such a point in a dedicated limitation section.

Answer: We agree with the reviewer that pharmacological treatments could interfere with non-cholesterol sterol concentrations. However, most studies were matched for medication use and we did not observe a systematic lack of effect in studies in which pharmacological treatments were given. Moreover, in the manuscript we discussed whether responses differed in medication-users compared to non-users (page 8, lines 199-201; page 17, lines 287-289).

Furthermore, the influence of diet should also be considered. Scicchitano P et al. (Journal of Functional Foods 2014;6:11-32) outlined the influence of nutraceuticals on lipid levels. Therefore, the need for a discussion about this point should also be targeted. Please provide.

Answer:Thank you for pointing this out and we agree that the influence of diet on lipid levels should also be considered. Therefore, we concluded in our manuscript that future research should confirm or refute our findings on cholesterol absorbers/synthesizers and the next step would be to use non-cholesterol sterol biomarkers for targeted (dietary) interventions. To emphasize the influence of diet, we added the reference of Scicchitano et al. as suggested by the reviewer (page 32, lines 462-465).

The authors should include a figure able to represent the entire results of this review in order to make it clearer to the Readers.

Answer:Thank you for this suggestion. Considering the many markers and metabolic disorders addressed, we believe that including an overview figure representing the entire results of the review would not improve the readability of the manuscript and will largely duplicate the findings from the tables. To summarize the main findings, we therefore decided to prepare table 9.

Reviewer 3 Report

The manuscript of Mashnafi et al. is a review of the literature on the inverse relationship in humans between cholesterol absorption and endogenous cholesterol synthesis in the context of a number of metabolic diseases, in total reviewing 94 studies.  The studies analyzed, as a mechanism for measuring the absorption, the efficacy of uptake of tracers to examine uptake of cholesterol and related molecules by intestinal epithelial cells (sitosterol, campesterol, cholestanol, lanthosterol, desmosterol).  I found the review a pleasure to read, with many important insights.  However, I have a few comments.

It was not made clear in the introduction about the relationship between NPC1L1 and ABCG5/G8.  The amount of cholesterol/sterol that accumulates in intestinal epithelial cells is dependent on the function and rate of activity of NPC1L1 pumping sterols into intestinal epithelial cells AND the function and rate of activity of ABCG5/G8 pumping sterols back into the intestinal lumen.  Since ABCG5/G8 could confound the perceived uptake rate of certain sterols (especially sitosterol if used as a tracer for uptake), then it is important to consider this.

As a continuation of the previous point, if NPC1L1 has a preferential specificity (or enhanced transfer) for certain sterols then using a certain tracer may cause over or underestimation of transport.  Likewise the specificity (or transfer activity) of ABCG5/G8 for certain sterols may cause an over or underestimation of transport.  Since NPC1L1 is regulated by SREBP2 and ABCG5/G8 by LXR, the activation of these transcription factors may also play a role.  The authors need to summarize how the previous papers have dealt with these issues in the introduction section.  This comment is highlighted by the fact that there are big differences in absolute values of measured sterol tracer uptake between studies.  

I find it fascinating that despite the big differences in plasma lipoprotein cholesterol in the different diseases (and therefore the mass of cholesterol that may reenter the liver), that there is not a regulation of cholesterol synthesis that is independent of dietary cholesterol uptake.  To be more specific: if plasma LDL-cholesterol is high and uptake of LDL-cholesterol in the liver is high, why don't we see a down-regulation of cholesterol synthesis independent of low cholesterol absorption?

What is the explanation for the high absorption/high synthesis phenotype in the deprived large intestine syndrome?  Is this the only disease where there isn't a reciprocal relationship?

Author Response

Reviewer #3

It was not made clear in the introduction about the relationship between NPC1L1 and ABCG5/G8.  The amount of cholesterol/sterol that accumulates in intestinal epithelial cells is dependent on the function and rate of activity of NPC1L1 pumping sterols into intestinal epithelial cells AND the function and rate of activity of ABCG5/G8 pumping sterols back into the intestinal lumen.  Since ABCG5/G8 could confound the perceived uptake rate of certain sterols (especially sitosterol if used as a tracer for uptake), then it is important to consider this.

As a continuation of the previous point, if NPC1L1 has a preferential specificity (or enhanced transfer) for certain sterols then using a certain tracer may cause over or underestimation of transport.  Likewise the specificity (or transfer activity) of ABCG5/G8 for certain sterols may cause an over or underestimation of transport.  Since NPC1L1 is regulated by SREBP2 and ABCG5/G8 by LXR, the activation of these transcription factors may also play a role.  The authors need to summarize how the previous papers have dealt with these issues in the introduction section.  This comment is highlighted by the fact that there are big differences in absolute values of measured sterol tracer uptake between studies.

          Answer:We agree with the reviewer in this important point. The two protein transporters NPC1L1 and ABCG5/G8 are involved in regulation of cholesterol and non-cholesterol sterols absorption in the intestinal lumen and indeed, we believe that the selectivity in sterol absorption is dependent on the actions of sterol influx (NPC1L1) and sterol efflux (ABCG5/G8) proteins. Therefore, genetic variation in these protein transporters may impact their activity, which could possibly have an effect on selectivity of cholesterol absorption markers; this has briefly been discussed in the manuscript (page 2, lines 83-86).

I find it fascinating that despite the big differences in plasma lipoprotein cholesterol in the different diseases (and therefore the mass of cholesterol that may re-enter the liver), that there is not a regulation of cholesterol synthesis that is independent of dietary cholesterol uptake. To be more specific: if plasma LDL-cholesterol is high and uptake of LDL-cholesterol in the liver is high, why don't we see a down-regulation of cholesterol synthesis independent of low cholesterol absorption?

Answer:Thank you for this intriguing remark. This is indeed a remarkable observation, for which there is currently no clear explanation. Normally the free intracellular cholesterol concentration is leading in cholesterol homeostasis and the serum LDL-cholesterol concentration is the consequence of these cellular processes, i.e. more or less receptor mediated uptake or transporter mediated secretion. In that perspective, it is remarkable that in conditions of high serum cholesterol concentrations together with downregulated endogenous synthesis, the intestine shows a high absorption. Although it is speculation, this might relate to the function of enterocytes, i.e. uptake of dietary fat-soluble vitamins and fatty acids, which depend on incorporation in chylomicrons, so there will be an ongoing chylomicron production that will always contain cholesterol. Due to the speculative nature of this suggestion, we did not incorporate this answer in the manuscript.

What is the explanation for the high absorption/high synthesis phenotype in the deprived large intestine syndrome?  Is this the only disease where there isn't a reciprocal relationship?

Answer: The observation that removal of colon disrupts this reciprocal relationship of cholesterol synthesis and absorption might suggest that absence of microflora could play a role. This suggestion has now been added in the manuscript (page 33, lines 512-514).

 Reviewer 4 Report

In this review article, the authors nicely summarized non-cholesterol sterol concentrations in different types of metabolic disorders. And the authors classified different situations based on such status as “cholesterol absorbers or cholesterol synthesizers”. This reviewer is very happy to see such nice summary, which could contribute to understand metabolic basis in different metabolic disorders. This reviewer would like to raises several suggestions to attract more readers, especially, physicians.

The authors classified different situations based on such status as “cholesterol absorbers or cholesterol synthesizers”. Is there any evidence suggesting that such situations affect the effectiveness of statins or ezetimibe? That would be very important for physicians to choose appropriate LDL-lowering therapy.

Is there any comments on several different specific situations caused by genetic, such as ABCG5/8 mutations, which could significantly affect on non-cholesterol sterol levels?

Author Response

Reviewer #4

The authors classified different situations based on such status as “cholesterol absorbers or cholesterol synthesizers”. Is there any evidence suggesting that such situations affect the effectiveness of statins or ezetimibe? That would be very important for physicians to choose appropriate LDL-lowering therapy.

Answer:The reviewer raised a valuable point here. It would indeed be important to know whether a subject can be classified as a cholesterol absorber or synthesizer to choose the most effective LDL-lowering therapy, e.g. ezetimibe for cholesterol absorbers and statins for cholesterol synthesizers. Therefore, we already concluded this by mentioning in the original manuscript that it is important to know this information for future (dietary) interventions. There is ample evidence, both from dietary interventions as well as pharmacological interventions, that effects of certain interventions are different in absorbers and synthesizers. We decided not to elaborate too extensively on this topic since that would almost be a manuscript on itself, but we did provide some examples of effectiveness of dietary and pharmacological intervention in cholesterol absorbers/synthesizers (page 32, lines 465-469).

Are there any comments on several different specific situations caused by genetic, such as ABCG5/8 mutations, which could significantly affect non-cholesterol sterol levels?

Answer:Again, many thanks for this suggestion. We have added a sentence stating that not only metabolic disorders but also genetic variation can affect non-cholesterol sterol concentrations (page 2, lines 83-86). 

Round  2

Reviewer 2 Report

To:

Editorial Board

Nutrients

Title: “Non-cholesterol sterol concentrations as biomarkers for cholesterol absorption and synthesis in different metabolic disorders: a systematic review”

 Dear Editor,

Please apologize the delay in answering.

I read the revised version of this manuscript and I think that the paper is good and well written.